# The Role of N^6^-Methyladenosine Modification in Microvascular Dysfunction

**DOI:** 10.3390/cells11203193

**Published:** 2022-10-11

**Authors:** Ye-Ran Zhang, Jiang-Dong Ji, Jia-Nan Wang, Ying Wang, Hong-Jing Zhu, Ru-Xu Sun, Qing-Huai Liu, Xue Chen

**Affiliations:** Department of Ophthalmology, The First Affiliated Hospital of Nanjing Medical University, Nanjing Medical University, Nanjing 210029, China

**Keywords:** N^6^-methyladenosine modification, angiogenesis, microvascular dysfunctions, epigenetics

## Abstract

Microvascular dysfunction (MVD) has long plagued the medical field despite improvements in its prevention, diagnosis, and intervention. Microvascular lesions from MVD increase with age and further lead to impaired microcirculation, target organ dysfunction, and a mass of microvascular complications, thus contributing to a heavy medical burden and rising disability rates. An up-to-date understanding of molecular mechanisms underlying MVD will facilitate discoveries of more effective therapeutic strategies. Recent advances in epigenetics have revealed that RNA methylation, an epigenetic modification, has a pivotal role in vascular events. The N^6^-methylation of adenosine (m^6^A) modification is the most prevalent internal RNA modification in eukaryotic cells, which regulates vascular transcripts through splicing, degradation, translation, as well as translocation, thus maintaining microvascular homeostasis. Conversely, the disruption of the m^6^A regulatory network will lead to MVD. Herein, we provide a review discussing how m^6^A methylation interacts with MVD. We also focus on alterations of the m^6^A regulatory network under pathological conditions. Finally, we highlight the value of m^6^A regulators as prognostic biomarkers and novel therapeutic targets, which might be a promising addition to clinical medicine.

## 1. Introduction

Microvascular dysfunction (MVD) remains a major health issue worldwide despite decades of research on its diagnosis, treatment, and prognosis. Featured by lesions in microvasculature, MVD leads to microvascular complications in various organs and systems [1,2]. Endothelial cells (ECs), pericytes, and vascular smooth muscle cells (VSMCs) are major components of microvasculature, whose proliferation, coverage, and dysfunction are key predictors of vascular fragility [3,4]. The etiology of MVD is heterogeneous and polymorphic. Various pathogenic factors, including hypoxia, inflammation, and metabolic disorders, contribute to MVD occurrence and development (Figure 1) [5].

RNA methylation is a group of epigenetic modifications that modulate gene expression without altering nucleotide sequences. RNA methylation includes 7-methylguanosine (m^7^G), 5-methylcytosine (m^5^C), 5-hydroxymethylcytosine (hm^5^C), N^1^-methyladenosine (m^1^A), N^6^-methyladenosine (m^6^A), N6,2′-O-dimethyladenosine (m^6^Am), and 2′-O′ methylation (2′-OMe). Among all, m^6^A modification is the most prevalent, abundant, and typical form in eukaryotes. Reportedly, m^6^A modification regulates the vascular regulatory network by mediating metabolism of vascular cells and expression of vascular genes [6,7]. Impaired m^6^A regulatory network disrupts microvascular homeostasis, further leading to MVD [8,9].

In this review, we summarized and discussed the role of m^6^A modification in MVD, aiming to provide a better understanding into its pathogenesis. Three dominant pathological processes of MVD were investigated, including neovascularization, microvascular malformation, and microvascular remodeling. This review also highlighted the potential clinical applications of m^6^A regulators as prognostic biomarkers and therapeutic targets for MVD.

## 2. RNA m^6^A Methylation

M^6^A modification, first detected in the 1970s, is the most abundant biochemical modification in eukaryotic RNAs, accounting for 0.1–0.4% of mammalian adenosine [10]. M^6^A modification has been identified in various types of RNAs, including messenger RNAs (mRNAs), transfer RNAs (tRNAs), ribosomal RNAs, long noncoding RNAs (lncRNAs), circular RNAs (circRNAs), small nuclear RNAs (snRNAs), and microRNAs (miRNAs). M^6^A modification participates in almost every step of RNA metabolism, from its generation, splicing, and processing in the nucleus to its translation, stabilization, and degradation in the cytoplasm, serving as a bridge between transcription and translation [11].

The global m^6^A level is dynamically regulated by writers and erasers, namely RNA methylases and demethylases respectively (Figure 2). M^6^A writers include methyltransferase-like 3/14/16 (METTL3/14/16), Wilms tumor 1-associated protein (WTAP), zinc finger CCCH-type containing 13 (ZC3H13), Vir-like m^6^A methyltransferase associated protein (VIRMA), and RNA-binding motif protein 15 (RBM15) [12]. The METTL3-METTL14 heterodimer and its catalytically inactive partner WTAP constitute the nucleus methyltransferase complex (MTC), which installs m^6^A modification. VIRMA, RBM15, and ZC3H13 are regulatory enzymes that facilitate recruitment of MTC [13]. RBM15 and ZC3H13 bind to the MTC and direct it to target RNA sites [14]. VIRMA regulates selective m^6^A methylation on 3′-UTR [14]. Reportedly, METTL16 is an independent writer that modifies snRNAs, U6 snRNA, and lncRNAs, but only a few substrates of METTL16 have been confirmed [14]. M^6^A erasers include fat mass and obesity-associated protein (FTO) and a-ketoglutarate-dependent dioxygenase alkB homolog 5 (ALKBH5). Both of them belong to the Fe2+/α-ketoglutarate-dependent dioxygenases enzyme family, which recognizes adenine and cytosine methylation in RNAs [14]. ALKBH5 also affects the synthesis and splicing of mRNAs [15]. RNA m^6^A sites are further recognized by m^6^A readers. Identified m^6^A readers include YT521-B homology (YTH) domain-containing proteins (YTHDF1/2/3, YTHDC1/2), insulin-like growth factor 2 mRNA-binding-proteins (IGF2BP1/2/3), heterogeneous nuclear ribonucleoprotein A2/B1 (hnRNPA2B1), and hnRNPC (Figure 2) [16]. hnRNPs and YTHDC1 are nuclear readers. hnRNPC binds to structurally altered RNAs and mediates pre-mRNA processing [14]. hnRNPA2B1 plays a vital role in RNA splicing and primary miRNA processing [14]. YTHDC1 mediates alternative splicing and facilitates mRNA export to cytoplasm [13]. In contrast, YTHDF1/2/3, YTHDC2, and IGF2BP1/2/3 are cytoplasmic-distributed. YTHDF1 recognizes m^6^A sites near the stop codon and enables mRNA translation by recruiting eukaryotic initiation factor 3, whereas YTHDF2 transports target mRNAs to the cytoplasmic processing body and promotes their degradation [17]. YTHDF3 is a modulator of YTHDF1 and YTHDF2, which can both enhance and suppress their effects [14]. The IGF2BP proteins are co-localized with Hu antigen R to enhance stability of target RNA transcripts [14]. They are also reported to participate in DNA replication and cell cycle progression [18].

## 3. M^6^A Modifications in Pathological Neovascularization

Neovascularization is defined as the sprouting of ECs in response to stimuli to form new capillary branches. The following steps are involved in neovascularization: (1) recognition of physiological or pathological signals, such as hypoxia, inflammation, and metabolic dysregulation; (2) secretion of proteases, pro-angiogenic factors and cytokines, and their bindings to corresponding receptors; (3) metabolic changes of vascular cells; (4) maturation of newly-formed vessels [19]. Reportedly, dysregulated epigenetic modifications, including DNA methylation, histone modifications, and RNA methylation, contribute to neovascularization [20]. Herein, we have summarized associations between aberrantly changed expression of m^6^A regulators and pathological neovascularization in Table 1.

### 3.1. M^6^A Modifications in Hypoxia-Related Neovascularization

Hypoxic effects are mediated by hypoxia-inducible factor (HIF), which combines with hypoxia-responsive elements (HREs) of target genes to regulate their expression [33]. There are three isoforms of HIF, including HIF-1, HIF-2, and HIF-3 [34]. HIFs are heterodimers composed of an α (HIF-1α, HIF-2α and HIF-3α) and a β (HIF-1β, HIF-2β and HIF-3β) subunit [35]. The C- and N-termini of α subunits have nuclear localization signals that direct them to nucleus to form adult HIFs [36]. Degradation of α subunits depends on prolyl hydroxylase domain-containing proteins (PHDs). Under normal conditions, PHDs target α subunits and mediate their polyubiquitination and degradation. However, activity of PHDs is disturbed upon hypoxia, thus interrupting the degradation of α subunits [37]. HIF-1α and HIF-2α share similar amino acid sequences and protein structures, and they regulate angiogenesis by targeting angiogenic factors (e.g., vascular endothelial growth factor (VEGF), angiopoietin-1/-2 (ANG-1/-2), transforming growth factor β (TGF-β), platelet-derived growth factor (PDGF)). However, the biological function of HIF-3 remains elusive [38,39].

Hypoxia could reprogram m^6^A epi-transcriptome, further reshaping downstream transcriptome and proteome that associate with neovascularization [40]. Increased METTL14 and ALKBH5 levels were detected in hypoxia-treated breast cancer cells, which led to upregulation of angiogenic transcripts, including TGF-β, matrix metallopeptidase 9 (MMP9), PDGF, and VEGFA. Conversely, METTL14/ALKBH5 knockdown reduced expression of angiogenic genes, thus inhibiting angiogenesis and cancer metastasis [23]. Hou et al. revealed a transcriptional inhibition of YTHDF2 by HIF-2α in hepatocellular carcinoma (HCC) cells. Suppressed expression of YTHDF2 not only promoted neovascularization through interleukin-11 (IL-11) and serpin family E member 2 (SERPINE2) but also led to microvascular malformation and remodeling. These adverse effects could be rescued by YTHDF2 upregulation [24]. Therefore, hypoxia primarily caused m^6^A changes, thus contributing to pathological neovascularization. Hypoxia-induced METTL3 downregulation in HCC promoted angiogenesis by upregulating expression of angiogenic genes, such as fibroblast growth factor, PDGF, and VEGFA, thus contributing to sorafenib resistance [25]. The Wnt signaling pathway is critical for vascular morphogenesis and endothelial specification [41]. Aberrantly activated Wnt signaling pathway is a leading cause of pathological neovascularization, particularly in wet age-related macular degeneration, diabetic retinopathy, and retinopathy of prematurity [42]. Yao et al. showed that METTL3 was upregulated in hypoxia-exposed retina [26]. METTL3 upregulation enhanced expression of LDL receptor related protein 6 (LRP6) and disheveled segment polarity protein 1 (DVL1) mRNAs, which promoted angiogenesis by activating Wnt signaling cascades [26].

Aberrantly changed expression of m^6^A regulators also facilitates HIFs generation and reprograms cellular metabolism, thus triggering neovascularization. In stomach cancer, IGF2BP3 directly targeted an m^6^A site in HIF-1α mRNA to upregulate its expression, leading to increased microvascular density and a poor outcome [22]. In HCC cells, METTL3, which was positively regulated by hepatitis B virus X-interacting protein (HBXIP), methylated HIF-1 mRNA to upregulate its expression, further contributing to the Warburg effect and angiogenesis [43]. Furthermore, in lung cancer, the crosstalk between polybromo 1 (PBRM1) and YTHDF2 was required for the effective synthesis of HIF-1 protein. YTHDF2 mediated RNA degradation in the cytoplasm under normal conditions, while it translocated into cell nucleus upon hypoxia to promote the cap-independent translation of HIF-1α mRNAs [21]. Collectively, these studies imply the critical role of m^6^A modification in hypoxia-induced neovascularization.

### 3.2. M^6^A Modifications in Inflammation-Related Pathological Neovascularization

Inflammation tends to induce irregularly shaped, leaky, and highly permeable angiogenesis rather than mature and functional vasculature [44]. Shan and colleagues detected altered expression of several m^6^A regulators, including FTO, METTL3, and METTL14, in mice with corneal neovascularization [27]. They further revealed that FTO promoted corneal neovascularization by inducing focal adhesion kinase (FAK) upregulation. In the alkali-burned corneal model, Yao et al. noticed that METTL3 knockdown restricted corneal neovascularization by inhibiting the Wnt pathway [26]. In HCC, YTHDF2 downregulation promoted neovascularization by accelerating the translation of inflammatory cytokines, such as IL-11 and SERPINE2 [24]. Similarly, lysine acetyltransferase 1 (KAT1) was poorly expressed in diabetic retinopathy, leading to YTHDF2 downregulation and inflammation-related neovascularization. YTHDF2 upregulation inhibited neovascularization and vascular leakage by degrading integrin subunit beta 1 (ITGB1) mRNAs and suppressing the FAK/PI3K/AKT signaling pathway [28]. These studies indicated the critical role of m^6^A modification in inflammation-related neovascularization.

### 3.3. Others

In this section, we present findings on the m^6^A-associated pathological angiogenesis in non-specific contexts. He et al. identified that decreased m^6^A level associated with reinforced angiogenesis and a poor survival rate in breast cancer [29]. They found that YTHDF3 promoted the binding between eukaryotic initiation factor 3 and angiogenic transcripts, such as VEGFA and epidermal growth factor receptor (EGFR), indicating its potential role as a therapeutic target in breast cancer [29]. Wang et al. found that METTL3 associated with angiogenesis and brain metastasis in lung cancer. Mechanistically, METTL3 promoted angiogenesis via facilitating the splicing of precursor miR-143-3p to generate its adult form, which positively regulated VEGFA expression [30]. Ma et al. identified miR-320b downregulation in lung cancer, which accelerated neovascularization through IGF2BP2-mediated thymidine kinase 1 (TK1) upregulation [45]. These results indicated that miRNAs and m^6^A regulators can be mutually regulated. In intrahepatic cholangiocarcinoma, FTO inhibited angiogenesis and tumor cell migration via upregulating C-C motif chemokine ligand 19 (CCL19) expression [31]. FTO also induced the apoptosis of intrahepatic cholangiocarcinoma cells by enhancing their sensitivity to cisplatin, indicating its potential role as a multipotent therapeutic target. In colorectal cancer/melanoma, ALKBH5 accelerated expression of angiogenic genes, such as VEGFA and TGFβ1, which weakened the efficacy of GVAX/anti–PD-1 therapy. These adverse effects could be rescued by the small-molecule ALKBH5 inhibitor (ALK-04) [32]. These studies revealed a critical role of m^6^A regulators in neovascularization and implied their potential therapeutic application in MVD.

## 4. M^6^A Modifications in Microvascular Malformation

Microvascular malformation mainly encompasses micro-venous malformation, arteriovenous malformation, lymphatic malformation, and mixed malformation [46]. Microvascular malformation, which can be congenital or acquired, arises from abnormal neovascularization, genetic mutations, and post-injury structural changes [46]. Endothelial dysplasia and incomplete pericyte coverage are two major characters of microvascular malformation [47]. Herein, we aim to discuss the association between m^6^A dysregulation and microvascular malformation (Table 2).

### 4.1. M^6^A Modifications in Hypoxia-Related Microvascular Malformation

M^6^A modification participates in hypoxia-related microvascular malformation by triggering incomplete pericyte coverage [48]. YTHDF2 positively regulates pericyte coverage by degrading m^6^A-containing IL-11 and SERPINE2 mRNAs [24]. YTHDF2 expression was suppressed in HIF-2α-treated HCC cells, which inhibited pericyte coverage and generated aberrant microvasculature. The HIF-2α blockade (PT2385) upregulated YTHDF2 expression, thus reversing the subsequent microvascular abnormalities in HCC [24]. Malignant tumors tend to obtain sufficient blood perfusion through vasculogenic mimicry, a vasculature-like structure formed by tumor cells instead of ECs [49]. Qiao and colleagues identified METTL3 upregulation in HCC, which facilitated both angiogenesis and vasculogenic mimicry [50]. Mechanistically, METTL3 aberrantly activated the Hippo pathway to generate vasculogenic mimicry, and upregulated angiogenic transcripts, such as vascular endothelial growth factor receptor 1/2 (VEGFR1/2) and matrix metallopeptidase 2/9 (MMP2/9), to promote angiogenesis [50]. Collectively, these studies implied a critical role of m^6^A modification in hypoxia-induced microvascular remodeling.

### 4.2. M^6^A Modifications in Inflammation-Related Microvascular Malformation

In diabetic retinopathy, METTL3 upregulation was detected in pericytes treated with inflammatory stimuli, such as tumor necrosis factor-α (TNF-α), and interleukin-6 (IL-6) [51]. METTL3 impaired viability, proliferation, and differentiation of pericytes via inhibiting the protein kinase C (PKC)/FAT4/PDGFRA axis in a YTHDF2-dependent manner. Conversely, Suo et al. detected that METTL3-specific deletion in pericytes promoted their coverage and suppressed diabetic microvascular complications [51]. In diabetic nephropathy, METTL14 was found to inhibit expression of α-klotho gene (an anti-inflammatory gene) and its encoding protein, leading to upregulation of inflammatory cytokines (TNF-α, IL-6) and microvascular malformation [52]. Therefore, a single m^6^A regulator may affect various downstream genes and set off a chain effect.

### 4.3. Others

Arteriovenous malformation is a vascular variation caused by the lack of capillary beds between venules and arterioles [53,54]. Wang et al. detected reduced METTL3 expression in arteriovenous malformation, which inhibited synergistic function of deltex E3 ubiquitin ligase 3L/1 (DTX1/3L) as Notch blockers, leading to aberrantly activated Notch signaling pathway and capillary malformation. These adverse effects could be restored by the Notch antagonist DAPT [55]. WTAP was also found downregulated in arteriovenous malformation, which caused capillary malformation through destabilizing desmoplakin (DSP), a critical component that maintains the integrity of vascular wall [56]. The Akt/mTOR signaling pathway is critical for endothelial differentiation [57]. In zebrafish embryos, METTL3 deletion in ECs upregulated the expression of PH domain and leucine rich repeat protein phosphatase 2 (PHLPP2), which promoted Akt dephosphorylation and suppressed the Akt/mTOR signaling pathway, thus leading to microvascular malformation [58]. Consistently, METTL3 deletion in bone mesenchymal stem cells also caused Akt dephosphorylation during osteogenic differentiation, thus inhibiting vascular normalization [59,60]. These microvascular defects were salvaged by Akt1 overexpression and/or the Akt activator SC79 [61]. These studies indicated a critical role of m^6^A modification in regulating Akt phosphorylation (Table 2).

## 5. M^6^A Modifications in Microvascular Remodeling

Microvascular remodeling is defined as structural or functional adaptations of the microvasculature. Either neovascularization or microvascular malformation can progress into microvascular remodeling (Figure 1) [62]. Herein, we have summarized associations between aberrantly changed expression of m^6^A regulators and microvascular remodeling in Table 3. 

### 5.1. M^6^A Modifications in Hypoxia-Related Microvascular Remodeling

Hypoxia-induced microvascular remodeling is primarily driven by HIF-2α [63]. Hou et al. identified that HIF-2α suppressed YTHDF2 expression in HCC. The reduced YTHDF2 level further provoked microvascular reconstruction by upregulating expression of IL-11 and SERPINE2 [24]. Pulmonary arterial hypertension is a lethal disease driven by progressive microvascular remodeling [64]. Proliferation of VSMCs is the main character of pulmonary arterial hypertension, manifested by concentric vasoconstriction and extracellular matrix deposition. METTL14 upregulation was observed in hypoxia-treated VSMCs, leading to progressive microvascular remodeling [65]. However, the downstream regulatory mechanism of METTL14-induced microvascular malformation remains elusive [65]. Proliferation of VSMCs depends on phosphatase and tensin homolog (PTEN), an endogenous inhibitor of PI3K/Akt/mTOR signaling cascades [66]. METTL3 upregulation in hypoxia-treated VSMCs mediated the degradation of PTEN mRNAs through YTHDF2 recognition. Thus, aberrant proliferation and migration of VSMCs occurred through Akt hyperphosphorylation, contributing to microvascular remodeling [67].

### 5.2. M^6^A Modifications in Inflammation-Related Microvascular Remodeling

Inflammation-related microvascular remodeling is driven by migration of inflammatory cells, which is mediated by adhesion molecules, such as intercellular adhesion molecule 1 (ICAM-1), vascular cell adhesion molecule 1 (VCAM-1), and E-selectin [62]. In atherosclerosis, METTL3 promoted microvascular remodeling by upregulating the expression of NLR family pyrin domain containing 1 (NRLP1), a gene generating inflammasomes, with YTHDF1 as the reader [68]. Moreover, METTL3 aggravated endothelial inflammation by inhibiting the expression of the anti-inflammatory protein KLF transcription factor 4 (KLF4) [68]. However, in TNF-α-treated ECs, METTL3 knockdown mitigated monocyte adhesion and microvascular remodeling [68]. In addition, METTL14 was also found upregulated in TNF-α-treated ECs, facilitating FOXO1 translation through YTHDF1 recognition [69]. FOXO1 then acted upon promoter regions of VCAM-1 and ICAM-1 mRNAs and promoted their transcriptions, contributing to microvascular remodeling.

### 5.3. M^6^A Modifications in Metabolism-Related Microvascular Remodeling

Metabolic disorders, such as dysregulation of glucose and lipid metabolism, also associate with microvascular remodeling [70]. VSMC dysfunction and intimal hyperplasia are two typical features of microvascular remodeling [71]. FTO upregulation in VSMCs was detected in type 2 diabetes mellitus, which triggered intimal hyperplasia through disturbing mRNA stability of smooth muscle 22 alpha (SM22α) [71]. YTHDC2 promoted circYTHDC2 expression in VSMCs under high glucose. CircYTHDC2 then inhibited the expression of ten-eleven translocation 2 (TET2), a gene positively regulating VSMC plasticity, thus contributing to VSMC dysfunction and microvascular remodeling. Metformin, a first-line hypoglycemic drug, alleviated YTHDF2-mediated microvascular remodeling by arresting cell cycle and inducing cell apoptosis [72,73].

Another leading cause of microvascular remodeling is dysregulated lipid metabolism. Macrophages take up oxidized lipoproteins and transform into foam cells, which cause endothelial dysfunction and extracellular matrix deposition, thus contributing to microvascular remodeling [74]. Gong et al. speculated that in atherosclerosis METTL14 promoted lncRNA ZFAS1 expression, an ncRNA that caused dyslipidemia. LncRNA ZFAS1 then elevated ADAM10/RAB22A expression to inhibit cholesterol efflux and facilitate microvascular remodeling [75]. The scavenger receptor CD36 is the primary transporter mediating lipid uptake and is directly targeted by PPARγ [76]. FTO inhibited foam cell formation by reducing CD36 and PPARγ levels. FTO also facilitated intracellular cholesterol efflux by upregulating ATP-binding cassette transporter A1 (ABCA1) expression, implying its potential role in preventing microvascular remodeling [76] (Table 3).

## 6. Discussion

MVD and its regulatory network have long been investigated. Various pathogenic factors, including hypoxia, inflammation, and metabolic disorders, contribute to MVD occurrence and development. RNA m^6^A modification is a post-transcriptional modification, which regulates all steps of RNA metabolism (splicing, maturation, export, translation, degradation). Herein, we summarized the role of m^6^A modification in MVD, aiming to provide a better understanding into its pathogenesis. M^6^A regulators participate in MVD pathogenesis by altering m^6^A status of vascular transcripts, thus mediating their expression. In turn, expression patterns of m^6^A regulators could also be changed by various pathogenic factors contributing to MVD. We also summarized the promising application of m^6^A modification in therapeutic strategies for MVD.

Roles and regulatory mechanisms of m^6^A regulators vary with their subcellular locations and in different diseases. Reportedly, stress induced the translocation of YTHDF2 from cytoplasm to nucleus, and unlike the role of cytoplasmic YTHDF2 in mediating RNA degradation, the endonuclear YTHDF2 promoted the cap-independent mRNA translation of HIF-1α, thus contributing to neovascularization [21,78]. M^6^A regulators may also play opposite roles in different diseases or pathogenesis. For instance, METTL3 promoted neovascularization in stomach cancer, but suppressed expression of angiogenic factors in sorafenib-resistant HCC [25,79]. FTO showed a pro-angiogenic role in diabetic retinopathy, but an anti-angiogenic role in intrahepatic cholangiocarcinoma [28,31]. The diversity is probably due to the distinct downstream regulatory network of METTL3 in different pathological processes. In addition, m^6^A modifications have been detected in various types of RNAs, while their roles in mediating metabolism of noncoding RNAs that associate with MVD are largely unknown. More investigations are warranted to reveal the complex biological/pathological effects and regulatory mechanisms of m^6^A modification.

Targeting m^6^A modification might be a promising therapeutic option for MVD. In colorectal cancer/melanoma, the ALKBH5 inhibitor ALK-04 downregulated expression of VEGFA and TGFβ1, thus inhibiting angiogenesis and enhancing efficacy of anti–PD-1 therapy [32]. Excitingly, in recent years, demethylation/methylation drugs, such as decitabine and azacitidine, have been developed, which have been clinically applied for the treatment of myelodysplastic syndrome and acute myeloid leukemia [80]. Both drugs are cytidine analogues that inhibit DNA methylation and restore normal function of tumor suppressor genes. Unlike decitabine, which only incorporates into DNA, azacitidine could be phosphorylated and incorporate into DNA/RNA, thus altering RNA synthesis and processing [81]. Reportedly, effects and sensitivities of antineoplastic drugs are enhanced by m^6^A regulators. In intrahepatic cholangiocarcinoma, FTO promoted cisplatin sensitivity to inhibit angiogenesis and accelerate the apoptosis of tumor cells [31]. ALKBH5 sensitized pancreatic ductal adenocarcinoma cells to gemcitabine by activating the Wnt pathway [82]. Moreover, in pancreatic cancer, suppressed METTL3 expression improved the efficacy of anti-cancer agents, such as gemcitabine, 5-fluorouracil, and cisplatin. These studies further suggested the potential clinical application of m^6^A modification in therapeutic strategies [83]. However, more investigations are needed to explore the role of m^6^A modification in MVD, thus helping with the development of prognostic and therapeutic strategies for MVD.

## Figures and Tables

**Figure 1 cells-11-03193-f001:**
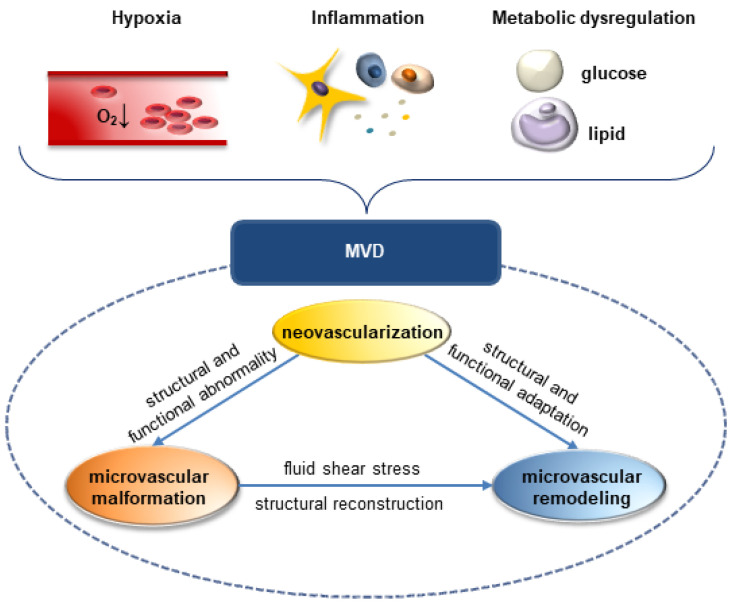
Pathogenic factors and pathological processes of MVD. Pathogenic factors, such as hypoxia, inflammation, and metabolic dysregulation, contribute to MVD. Pathological processes of MVD include neovascularization, microvascular malformation, and microvascular remodeling. Both neovascularization and microvascular malformation can be structurally and functionally remodeled in response to physical and chemical stimuli.

**Figure 2 cells-11-03193-f002:**
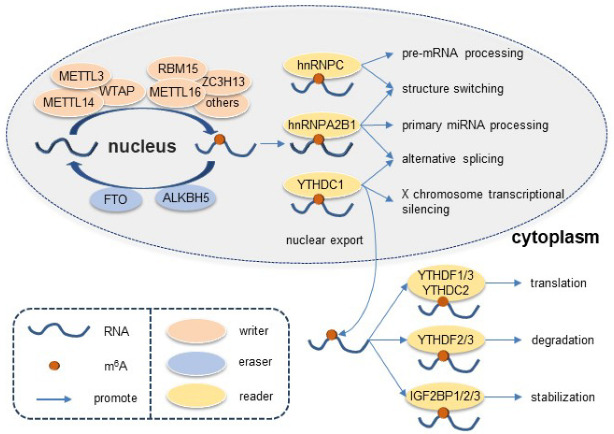
The process and molecular functions of RNA m^6^A methylation. M^6^A modification is dynamically installed by writers (METTL3, METTL4, METTL14, WTAP, RBM15, ZC3H13) and removed by erasers (FTO and ALKBH5). M^6^A sites are recognized by readers. hnRNPs and YTHDC1 are nuclear readers. hnRNPC binds to structurally altered RNAs and mediates pre-mRNA processing. hnRNPA2B1 regulates RNA splicing and primary miRNA processing. YTHDC1 mediates alternative splicing and facilitates mRNA export to cytoplasm. YTHDF1/2/3, YTHDC2, and IGF2BP1/2/3 are cytoplasmic-distributed. YTHDF1 enables mRNA translation by recruiting eukaryotic initiation factor 3, whereas YTHDF2 transports target mRNAs to the cytoplasmic processing body and promotes their degradation. YTHDF3 is a modulator of YTHDF1 and YTHDF2, which can both enhance and suppress their effects. IGF2BPs enhance stability of target RNA transcripts.

**Table 1 cells-11-03193-t001:** Molecular mechanisms of m^6^A modification in pathological neovascularization.

Pathological Process	Disease	M^6^A Regulators	Model System			Mechanism	Reference
			Human Tissue	Animal Model	Cell Line		
hypoxia	lung cancer	YTHDF2↑	√		√	promote HIF-1 expression	[21]
	stomach cancer	IGF2BP3↑	√		√	promote HIF-1 expression	[22]
	breast cancer	METTL14 /ALKBH5↑	√	√	√	increase TGFβ1 expression	[23]
	HCC	YTHDF2↓	√	√	√	stabilize IL-11 and SERPINE2 mRNA	[24]
		METTL3↓	√	√	√	increase PDGF and VEGF expression	[25]
	oxygen-induced retinopathy	METTL3↑		√	√	activate the Wnt pathway	[26]
inflammation	HCC	YTHDF2↓	√	√	√	stabilize IL-11 and SERPINE2 mRNA	[24]
	corneal neovascularization	FTO↑		√	√	increase FAK expression	[27]
		METTL3↑		√	√	activate the Wnt signaling pathway	[26]
	diabetic retinopathy	YTHDF2↓		√	√	activate FAK/PI3K/AKT pathway	[28]
others	breast cancer	YTHDF3↑	√	√	√	enhance translation of VEGF	[29]
	lung cancer	METTL3↑	√	√	√	increase VEGFA expression	[30]
	intrahepatic cholangiocarcinoma	FTO↓	√	√	√	increase CCL19 expression	[31]
	colorectal cancer/melanoma	ALKBH5↑	√	√	√	promote VEGF expression	[32]

Abbreviations: HCC, human hepatocellular carcinoma; IL-11, interleukin-11; SERPINE2, serpin family E member 2; FAK, focal adhesion kinase; VEGF, vascular endothelial growth factor; TGF-β, transforming growth factor β; CCL19, C-C motif chemokine ligand 19; ↑, upregulation; ↓, downregulation; √, the experimental model was included.

**Table 2 cells-11-03193-t002:** Molecular mechanisms of m^6^A modification in microvascular malformation.

Pathological Process	Disease	M^6^A Regulators	Model System			Mechanism	Reference
			Human Tissue	Animal Model	Cell Line		
hypoxia	HCC	YTHDF2↓	√	√	√	stabilize IL-11 and SERPINE2 mRNA	[24]
		METTL3↑	√	√	√	activate Hippo pathway	[50]
inflammation	diabetic nephropathy	METTL14↑	√	√	√	decrease α-klotho expression	[52]
	diabetic retinopathy	METTL3↑		√	√	suppress PKC/FAT4/PDGFRA pathway	[51]
others	arteriovenous malformation	METTL3↓	√		√	activate the Notch pathway	[55]
		WTAP↓	√		√	block the Wnt pathway	[56]
	model system (endothelial cells)	METTL3↓		√	√	inhibit the PI3K/AKT pathway	[58]
	model system (bone mesenchymal stem cells)	METTL3↓			√	inhibit the PI3K/AKT pathway	[59,60]

Abbreviations: ↑, upregulation; ↓, downregulation; √, the experimental model was included.

**Table 3 cells-11-03193-t003:** Molecular mechanisms of m^6^A modification in microvascular remodeling.

Pathological Process	Disease	M^6^A Regulators	Model System			Mechanism	Reference
			Human Tissue	Animal Model	Cell Line		
hypoxia	HCC	YTHDF2↓	√	√	√	stabilize IL-11 and SERPINE2 mRNA	[24]
	pulmonary arterial hypertension	METTL3↑		√	√	degrade PETN mRNAs	[67]
		METTL14↑		√		cooperate with SETD2	[65]
inflammation	atherosclerosis	METTL3↑		√	√	increase NLRP1 and decrease KLF4 expression	[68]
		METTL14↑		√	√	increase VCAM-A and ICAM-1 expression	[69,77]
metabolism	type 2 diabetes mellitus	FTO↑		√	√	destabilize SM22α mRNAs	[71]
		YTHDC2↑		√	√	inhibit TET2 expression	[72]
	atherosclerosis	FTO↑	√	√	√	reduce CD36 and PPARγ level	[76]

Abbreviations: NRLP1, NLR family pyrin domain containing 1; KLF4, KLF transcription factor 4; ICAM-1, intercellular adhesion molecule 1; VCAM-1, vascular cell adhesion molecule 1; SM22α, smooth muscle 22 alpha; PTEN, phosphatase and tensin homolog; ABCA1, ATP-binding cassette transporter A1; CD36, CD36 molecule; ↑, upregulation; ↓, downregulation; √, the experimental model was included.

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
