# Peer review of "The Role of N6-Methyladenosine Modification in Microvascular Dysfunction"

_cells, 2022, doi:10.3390/cells11203193_

Round 1

Reviewer 1 Report

The review article by Zhang et al. on the role of N6 methyladenosine in microvascular biology is very complete and interesting.  I have only a few concerns that affect the readability of the work.  They are all relatively minor however.

1.     Table 1 is too far away from its first reference.  It should be moved higher in the text.  Within 1 to 1 ½ pages from first reference to it.

2.     Table 1 could be vastly improved columns 4-7 repetitively (but also somewhat cryptically) say the same thing.  As an example, the 6th row on Fox O3.  METTL3/YTHDF2 are downregulated and this promotes FoxO3 degradation.  Column 4 could be removed by just putting a down arrow by METTL3/YTHDF2.  And then remove column 5 and 6, and just have the “mechanism” column that says “promotes FOXO3 mRNA degradation”.  This would reduce 5 columns into 2. 

3.     The initial used in table 1 are never defined in the table and one has to search through the text to figure out what they mean.  If the above reduction in columns is done, then HCC, ICC, LC, SC, could all be written out as full words.  Alternatively, a post-script under the table should define them all.

4.     In table 1, HUVECs are not a disease.  In fact, most entries in that column are cell types rather than diseases.  It could be renamed “model system” or something else.

5.     Lastly for table 1, Mechanism and References should have the first letter capitalized.

6.     There are far too many acronyms for easy readability.  CDS/WSS are only mentioned when they are defined and then they never come up again.  VM is only mentioned in one paragraph, it can be written out in full every time. DN is mentioned one when it is defined, once in the table and only one time after that. DR is only mentioned when it is defined and in the table.  The same for many other (LC, BMSC, etc.).  BC is defined, and then not used again for pages (at which point, I had forgotten what it stood for).  I spent a lot of time reading this review chasing the definition of acronyms.

7.     HCC was not defined on first use but on second or third use. 

8.     A few situations, the grammar reduces readability:

-       Page 3, “the alpha subunit is then translocated into”.  The use of a passive sentence here is weird.  “The alpha subunit translocats into..”

-       On line 156, “it’s likely to speculate that m6A..” First, “it’s” is too informal for a review. There is also a second location where “it’s” is used and both should be changed to “it is”.  But in this case, also the sentence is strange.  Either it is speculated or it is not.  “It is likely to speculate” is strange.  Perhaps the authors meant “ It can be speculated that… “

-       On line 355, “ are an interconnected cellular signaling that..” is missing a word.  Maybe “interconnected signaling pathway that ..”

-       On line 357, the sentence also ends abruptly such that it feels like something is missing.  “The METTL3/PHLPP2/mTOR-Akt axis is one of the regulatory mechanisms”.. that does what?  That is important for vascular physiology? 

9.     On page 4, the sentence “hypoxia reprograms the m6A epi-transcriptome” appears in very similar wording on line 132 and 136.  Being so close to each other, it reads very repetitive.

10. On line 165, the sentence “The Wnt system.. ” should be used to start a new paragraph.

11. On line 233, the section subtitle is on one page, alone, with the text starting on the next page.  The journal does not do formatting for the authors, so I would suggest the authors fix this.

Author Response

Response to Reviewer 1 Comments

  1. Table 1 is too far away from its first reference.  It should be moved higher in the text.  Within 1 to 1 ½ pages from first reference to it.

Response 1: Thank you for your comment. As suggested by the reviewer, we have replaced the Table 1 to make it closer to the first reference. We hope the reviewer will find it appropriate now.

  1. Table 1 could be vastly improved columns 4-7 repetitively (but also somewhat cryptically) say the same thing.  As an example, the 6th row on FoxO3.  METTL3/YTHDF2 are downregulated and this promotes FoxO3 degradation. Column 4 could be removed by just putting a down arrow by METTL3/YTHDF2. And then remove column 5 and 6, and just have the “mechanism” column that says “promotes FOXO3 mRNA degradation”.  This would reduce 5 columns into 2. 
  2. The initial used in table 1 are never defined in the table and one has to search through the text to figure out what they mean. If the above reduction in columns is done, then HCC, ICC, LC, SC, could all be written out as full words. Alternatively, a post-script under the table should define them all.
  3. In table 1, HUVECs are not a disease.  In fact, most entries in that column are cell types rather than diseases.  It could be renamed “model system” or something else.
  4. Lastly for table 1, Mechanism and References should have the first letter capitalized.

Responses 2-5: Thank you so much for your comment. Since comments 2-5 are all on the structures and contents of Table 1, we therefore have merged our responses into one. The second reviewer also raised comments on the tables. We therefore have modified all three Tables according to suggestions by both reviewers. Please find the below Tables 1-3. We hope the reviewer will find it appropriate now.

Table 1.  Molecular mechanisms of m6A modification in pathological neovascularization

Pathological process

Disease

M6A regulators

Model system

Mechanism

Human tissue

Animal model

Cell line

hypoxia

lung cancer

YTHDF2↑

promote HIF-1 expression

stomach cancer

IGF2BP3↑

promote HIF-1 expression

breast cancer

METTL14

/ALKBH5↑

increase TGFβ1 expression

HCC

YTHDF2↓

stabilize IL-11 and SERPINE2 mRNA

METTL3↓

increase PDGF and VEGF expression

oxygen-induced retinopathy

METTL3↑

activate the Wnt pathway

inflammation

HCC

YTHDF2↓

stabilize IL-11 and SERPINE2 mRNA

corneal neovascularization

FTO↑

increase FAK expression

METTL3↑

activate the Wnt signaling pathway

diabetic retinopathy

YTHDF2↓

activate FAK/PI3K/AKT pathway

others

breast cancer

YTHDF3↑

enhance translation of VEGF

lung cancer

METTL3↑

increase VEGFA expression

intrahepatic cholangiocarcinoma

FTO↓

increase CCL19 expression

colorectal cancer / melanoma

ALKBH5↑

promote VEGF expression

Abbreviations: HCC, human hepatocellular carcinoma; IL-11, interleukin-11; SERPINE2, serpin family E member 2; FAK, focal adhesion kinase; VEGF, vascular endothelial growth factor; TGF-β, transforming growth factor β; CCL19, C-C motif chemokine ligand 19;

Table 2.  Molecular mechanisms of m6A modification in microvascular malformation

Pathological process

Disease

M6A regulators

Model system

Mechanism

Human tissue

Animal model

Cell line

hypoxia

HCC

YTHDF2↓

stabilize IL-11 and SERPINE2 mRNA

METTL3↑

activate Hippo pathway

inflammation

diabetic nephropathy

METTL14↑

decrease α-klotho expression

diabetic retinopathy

METTL3↑

suppress PKC/FAT4/PDGFRA pathway

others

arteriovenous malformation

METTL3↓

activate the Notch pathway

WTAP↓

block the Wnt pathway

model system (endothelial cells)

METTL3↓

inhibit PI3K/AKT pathway

model system (bone mesenchymal stem cells)

METTL3↓

inhibit PI3K/AKT pathway

Table 3. Molecular mechanisms of m6A modification in microvascular remodeling

Pathological process

Disease

M6A regulators

Model system

Mechanism

Human tissue

Animal model

Cell line

hypoxia

HCC

YTHDF2↓

stabilize IL-11 and SERPINE2 mRNA

pulmonary arterial hypertension

METTL3↑

degrade PETN mRNAs

METTL14↑

cooperate with SETD2

inflammation

atherosclerosis

METTL3↑

increase NLRP1 and decrease KLF4 expression

METTL14↑

increase VCAM-A and ICAM-1 expression

metabolism

type 2 diabetes mellitus

FTO↑

destabilize SM22α mRNAs

YTHDC2↑

inhibit TET2 expression

atherosclerosis

FTO↑

reduce CD36 and PPARγ level

Abbreviations: NRLP1, NLR family pyrin domain containing 1; KLF4, KLF transcription factor 4; ICAM-1, intercellular adhesion molecule 1; VCAM-1, vascular cell adhesion molecule 1; SM22α, smooth muscle 22 alpha; PTEN, phosphatase and tensin homolog; ABCA1, ATP-binding cassette transporter A1; CD36, CD36 molecule;

  1. There are far too many acronyms for easy readability. CDS/WSS are only mentioned when they are defined and then they never come up again. VM is only mentioned in one paragraph, it can be written out in full every time. DN is mentioned one when it is defined, once in the table and only one time after that. DR is only mentioned when it is defined and in the table.  The same for many other (LC, BMSC, etc.). BC is defined, and then not used again for pages (at which point, I had forgotten what it stood for). I spent a lot of time reading this review chasing the definition of acronyms.

Response 6: Thank you for your comment. We are sorry for the confusion caused. We have modified acronyms throughout the manuscript according to the reviewer’s suggestion. We hope the reviewer will find it appropriate now.

  1. HCC was not defined on first use but on second or third use. 

Response 7: Thank you for your comment. We have modified the draft accordingly. We hope the reviewer will find it appropriate now.

  1. A few situations, the grammar reduces readability:

- Page 3, “the alpha subunit is then translocated into”. The use of a passive sentence here is weird. “The alpha subunit translocats into..”

-On line 156, “it’s likely to speculate that m6A…” First, “it’s” is too informal for a review. There is also a second location where “it’s” is used and both should be changed to “it is”. But in this case, also the sentence is strange. Either it is speculated or it is not. “It is likely to speculate” is strange. Perhaps the authors meant “ It can be speculated that… “

-On line 355, “ are an interconnected cellular signaling that..” is missing a word. Maybe “interconnected signaling pathway that ..”

-On line 357, the sentence also ends abruptly such that it feels like something is missing. “The METTL3/PHLPP2/mTOR-Akt axis is one of the regulatory mechanisms” that does what?  That is important for vascular physiology? 

Response 8: Thank you for your comment. We accept all above suggestions from reviewer and have the English language checked by a native speaker. Below are our specific modifications:

(1) Line 132: The C- and N-termini of α subunits have nuclear localization signals that direct them to nucleus to form adult HIFs. Degradation of α subunits depends on prolyl hydroxylase domain-containing proteins (PHDs). Under normal conditions, PHDs target α subunits and mediate their polyubiquitination and degradation.

(2) Line 149-151: Hypoxia-induced METTL3 downregulation in HCC promoted angiogenesis by upregulating expression of angiogenic genes, such as fibroblast growth factor, PDGF, and VEGFA, thus contributing to sorafenib resistance.

(3) Line 255: The Akt/mTOR signaling pathway is critical for endothelial differentiation.

(4) Line 255-259: In zebrafish embryos, METTL3 deletion in ECs upregulated the expression of PH domain and leucine rich repeat protein phosphatase 2 (PHLPP2), which promoted Akt dephosphorylation and suppressed the Akt/mTOR signaling pathway, thus leading to microvascular malformation.

We hope you will find it appropriate now.

  1. On page 4, the sentence “hypoxia reprograms the m6A epi-transcriptome” appears in very similar wording on line 132 and 136. Being so close to each other, it reads very repetitive.

Response 9: Thank you for your comment. We have modified the text as below to avoid repetition:

“HIF-1α and HIF-2α share similar amino acid sequences and protein structures, and they regulate angiogenesis by targeting angiogenic factors (e.g. vascular endothelial growth factor (VEGF), angiopoietin-1/-2 (ANG-1/-2), transforming growth factor β (TGF-β), platelet-derived growth factor (PDGF)). However, the biological function of HIF-3 remains elusive. Hypoxia could reprogram m6A epi-transcriptome, further reshaping downstream transcriptome and proteome.”

We hope you will find it appropriate now.

  1. On line 165, the sentence “The Wnt system” should be used to start a new paragraph.

Response 10: Thank you for your comment. We have adjusted the overall structure of the review. In the current version, the Wnt section only discusses findings in one published paper, we do not see the need for it to be a separate paragraph. We hope you will find it appropriate now.

  1. On line 233, the section subtitle is on one page, alone, with the text starting on the next page.  The journal does not do formatting for the authors, so I would suggest the authors fix this.

Response 11: Thank you for your comment. We have reformatted the text according.

Reviewer 2 Report

The manuscript "Novel Sights Into The Role Of N6-methyladenosine Modification In Microvascular Dysfunction" by Ye-Ran Zhang and co-authors reviews the current literature regarding m6A in microvascular dysfunction focusing on hypoxia- and inflammation-related pathologies. The review provides a good overview of the current (very limited) understanding and addresses the discrepancies in the field. The review can be expected to be very helpful for researchers of the field or entering the field as a resource collecting research in this area.

Specific concerns:

--The review article format is intended to provide insight into current knowledge. Not to provide "novel sights" as would be the case for a perspective article. It is imperative that the authors change the title and avoid use of words such as new or novel. The title must reflect the content, which is a review of the current knowledge based on literature.

--Microvascular dysfunction is a description of an overall pathology incorporating several types of mechanisms. It would make sense to use it as a singular, thus MVD and not MVDs.

--Lines 34-36:Vascular fragility can also be considered to be due to vascular cell components functional disruptions, not just proliferation and coverage. Please revise.

--Line 73: Be clear already in the list to separate METTL3/14 complex and its catalytically inactive partners as well as METTL16

--Line 84: The readers do not bind m6A but specific RNA sequences containing m6A

--Lines 96-97: This is an oversimplification and as such incorrect. Not enough is known about the readers. The current understanding is that YTHDF3 is a modulator of YTHDF1 and YTHDF2 actions. It has been shown to both enhance and suppress their actions. Please see e.g. https://pubmed.ncbi.nlm.nih.gov/33295243/

--Line 125: The authors point to the existence of different HIFs. Please elaborate on the different isoforms.

--Lines136-140: It is unclear if the authors consider changed m6A to be a secondary response to hypoxia-mediated changes in the cells or if they consider hypoxia primarily to cause m6A-changes. Please clarify the causality.

--The authors review the current literature and compile the most important findings in tables. The tables are very important and should incorporate an evaluation of the strength of each cited result. It is especially important to note in the table if the data derives from human or experimental tissues, animal models or cell line studies.

--The English language must be checked by a professional linguistic editor native in English language.